# Rational development of catalytic Au(I)/Au(III) arylation involving mild oxidative addition of aryl halides

Abdallah Zeineddine[1], Laura Estévez[2,3], Sonia Mallet-Ladeira[4], Karinne Miqueu[2], Abderrahmane Amgoune [1] & Didier Bourissou [1]

The reluctance of gold to achieve oxidative addition reaction is considered as an intrinsic limitation for the development of gold-catalyzed cross-coupling reactions with simple and ubiquitous aryl halide electrophiles. Here, we report the rational construction of a Au(I)/Au(III) catalytic cycle involving a sequence of $Csp^2$–X oxidative addition, $Csp^2$–H auration and reductive elimination, allowing a gold-catalyzed direct arylation of arenes with aryl halides. Key to this discovery is the use of Me-Dalphos, a simple ancillary (P,N) ligand, that allows the bottleneck oxidative addition of aryl iodides and bromides to readily proceed under mild conditions. The hemilabile character of the amino group plays a crucial role in this transformation, as substantiated by density functional theory calculations.

[1] CNRS, Université Paul Sabatier, Laboratoire Hétérochimie Fondamentale et Appliquée (LHFA, UMR 5069), 118 Route de Narbonne, 31062 Toulouse Cedex 09, France. [2] CNRS/UNIV PAU & PAYS ADOUR, Institut des Sciences Analytiques et de Physico-Chimie pour l'environnement et les Matériaux (IPREM UMR 5254), Hélioparc, 2 avenue du Président Angot, 64053 PAU Cedex 09, France. [3] Departamento de Química Física, Universidade de Vigo, Facultade de Química Lagoas-Marcosende s/n, Vigo, Galicia 36310, Spain. [4] Institut de Chimie de Toulouse (FR 2599), 118 Route de Narbonne, 31062 Toulouse Cedex 09, France. Correspondence and requests for materials should be addressed to A.A. (email: amgoune@chimie.ups-tlse.fr) or to D.B. (email: dbouriss@chimie.ups-tlse.fr)

Transition metal catalysis has spectacularly progressed over the past century and nowadays occupies a forefront position in organic synthesis. The case of gold is particularly fascinating. It was long considered as too inert and thus useless synthetically. However, the situation has changed dramatically and gold catalysis has become a very active and flourishing field[1]. As far as homogeneous catalysis is concerned, gold complexes are now considered to be the most efficient and versatile catalysts for the electrophilic activation of carbon–carbon π-bonds. A number of useful catalytic transformations based on π- or σ,π-coordination of alkynes, allenes and alkenes have been developed[2].

Comparatively, cross-coupling reactions, which are arguably the most important transformations to build carbon–carbon and carbon–heteroelement bonds, and are well known with about all mid and late transition metals (in particular, Pd, Rh, Cu, Ni, Fe)[3], have only recently emerged in the gold portfolio[4–6]. The paucity of gold-catalyzed cross-coupling reactions has to do with the reluctance of gold to cycle between its I and III oxidation states. Oxidative addition, which is the common entry point to cross-coupling catalytic cycles, is not favored for Au(I) complexes[7–11]. Recent studies have shown that this limitation can be circumvented using strong external oxidants such as hypervalent iodine or electrophilic fluorinating reagents, which give access to the Au(III) oxidation state and enable turnover in so-called oxidative cross-couplings[12–20]. Alternatively, strong electrophiles such as aryl diazonium or diaryliodonium salts can be used to oxidize gold under thermal or photochemical conditions[21–23]. This approach has been successfully applied to the cross-coupling of aryl diazonium salts with aryl boronic acids or alkynes merging photoredox and gold catalysis[24–28].

The reluctance of gold to undergo oxidative addition presents some advantages synthetically (orthogonal functional group tolerance with transition metals commonly used in cross-couplings), but it severely limits the scope of possible transformations. Clearly, oxidative addition step is the bottleneck to achieve and develop cross-coupling reactions with simple and ubiquitous electrophiles such as aryl halides. On the other hand, some features make gold quite unique and very attractive for cross-coupling reactions. Indeed, thanks to their high electrophilic character, Au(III) complexes are known to readily activate $Csp^2$–H and even $Csp^3$–H bonds under mild conditions[29–31], and they have garnered heightened interest in cross-coupling reactions involving C–H bond activation[12–20]. Thus, there is great interest and significance in triggering and controlling the reactivity of Au(I) complexes to achieve oxidative addition of aryl halides under mild conditions. Combined with the ability of Au(III) species for $Csp^2$–H bond activation of (hetero)arenes and the easiness of $Csp^2$–$Csp^2$ reductive elimination at gold[31, 32], this may open the way to gold-catalyzed coupling reactions with aryl halides via Au(I)/Au(III) redox cycles.

The reluctance of Au(I) complexes toward oxidative addition is generally attributed to the high redox potential of the Au(I)/Au(III) couple (1.41 V) compared to those of Pd(0)/Pd(II) (0.91 V) or Pt(0)/Pt(II) (1.18 V)[33]. However, a few recent studies have challenged this paradigm and some Au(I) complexes have been shown to indeed undergo oxidative addition (Fig. 1a)[31, 34–38]. In particular, we have reported that cationic Au(I) complexes featuring a small bite angle diphosphino-carborane (DPCb) ligand readily achieve the oxidative addition of $Csp^2$–I and strained C–C bonds[37, 38]. However, the strong preference of Au(I) for two-coordinate linear geometry[7, 39, 40] considerably limits ligand modulation. Indeed, it is difficult to chelate Au(I) with small bite angle ligands. In most cases, Au⋯Au dinuclear structures are obtained instead[41]. Au(I) cationic complexes featuring monodentate ligands (L=$R_3P$ or N-heterocyclic carbenes) have also been shown to undergo oxidative addition[35, 42, 43], but the

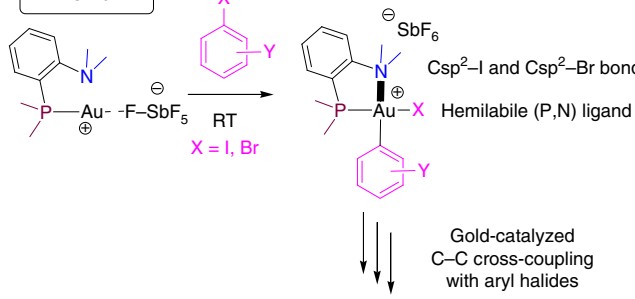

**Fig. 1** Intermolecular oxidative additions to Au(I) complexes. **a** Recent examples of stoichiometric oxidative addition to gold. **b** This work: mild oxidative addition of aryl halides to gold using a hemilabile (P,N) ligand and gold-catalyzed arylation with aryl halides

generated three-coordinate Au(III) species are highly unstable and difficult to exploit in further reactivity. In contrast, four-coordinate Au(III) complexes are quite stable[44]. We thus hypothesized that hemilabile bidentate ligands could provide a suitable balance between reactivity and stability of the key Au(III) species. Taking advantage of the soft/hard character of Au(I)/Au(III), we envisioned to use (P,N) bidentate ligands such as [$R_2P$($o$-$C_6H_4$)$NR'_2$][45–47]. The $κ^1$–P coordination to the soft Au(I) center would provide a reactive gold cationic complex, and upon oxidative addition, the pendant amine group would coordinate the resulting hard Au(III) center and temper its reactivity[21, 48].

Here, we show that this ligand design principle is valid. Easily accessible cationic Au(I) complexes featuring a hemilabile (P,N) bidentate ligand are shown to readily promote oxidative addition of a large scope of aryl iodides and bromides. The potential of (P,N) Au(I) complexes in Au(I)/Au(III) catalysis is also demonstrated by the development of arylation reactions involving aryl halides and catalyzed by well-defined mononuclear gold complexes. Precedents of gold-catalyzed cross-coupling reactions with aryl halides are extremely rare[49–51], and involve polynuclear species.

## Results

**Oxidative addition of PhI to (Me-Dalphos) gold complexes**. To start with, we studied the reactivity of the commercially available (Me-Dalphos) AuCl complex **1** toward iodobenzene in the presence of AgNTf$_2$ (Fig. 2a). Gratifyingly, $^{31}$P nuclear magnetic resonance (NMR) monitoring indicated gradual conversion of the cationic complex into the (Me-Dalphos) Au(III)–aryl complex **2** (δ74.1 p.p.m.) as the major phosphorus containing compound, along with small amount of unidentified species. The reaction is

**a**

**b**

**Fig. 2** Generation of the Me-Dalphos gold(III) phenyl complex **2**. **a** Oxidative addition of iodobenzene to complex **1**. **b** Molecular structure of complex **2**, hydrogen atoms and the NTf$_2$ counter-anion are omitted for clarity, selected bond lengths (Å) and angles (°): Au-N 2.214(5), Au-P 2.369(2), Au-C 2.064 (6), Au-I 2.632(1) and P-Au-N: 84.62(13)

**Fig. 3** Oxidative addition of biphenylene to gold. Reaction of biphenylene with complex **1** in the presence of AgSbF$_6$ (Ad = 1-adamantyl)

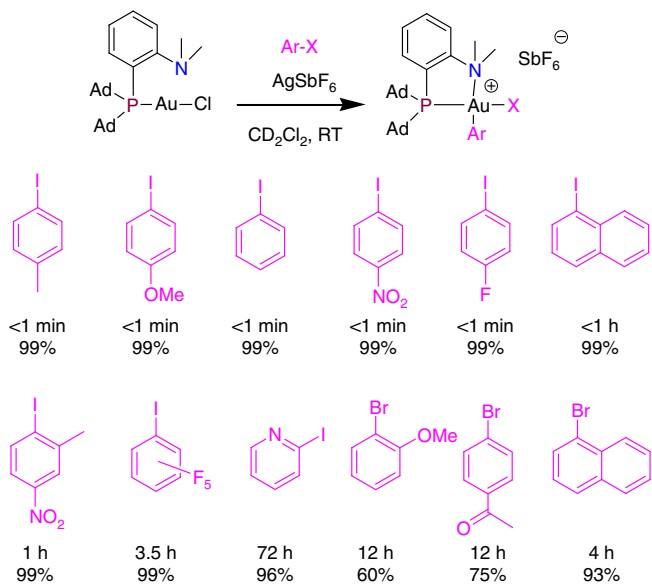

**Fig. 4** Oxidative addition of aryl iodides and bromides to gold. Reaction of a series of aryl iodides and bromides with complex **1** in the presence of AgSbF$_6$. Reaction times to reach complete conversion and spectroscopic yields are based on $^{31}$P NMR

complete after 48 h at room temperature (**2** is formed in 92% spectroscopic yield). Most diagnostic in NMR is the downfield shift of the $^{13}$C NMR signal for the quaternary carbon atom, from 92.6 p.p.m. in PhI to 127.7 p.p.m. in the Au(III)–Ph complex **2**[37], [44]. The latter signal appears as a broad singlet without any observable C–P coupling constant, suggesting that the phenyl ring sits in *cis* position to phosphorus. Concomitantly, the $^1$H NMR resonance signal for the N(CH$_3$)$_2$ substituent is very deshielded at δ3.50 p.p.m., indicating that the nitrogen atom is coordinated to the Au(III) center. Crystals of **2** were obtained from a concentrated dichloromethane solution. The X-ray diffraction analysis (Fig. 2b) confirmed κ$^2$–(P,N) coordination of the

Me-Dalphos ligand (d(Au–N)=2.214(5) Å, P–Au–N bite angle=84.62(13)°). Complex **2** displays a discrete ion pair structure. The gold center is tetracoordinate and adopts square-planar geometry. In line with that observed in solution, the phenyl ring sits in *cis* position to phosphorus, so as to form the most stable stereoisomer (nitrogen exerts a significantly weaker *trans* influence than phosphorus).

Interestingly, the counter-anion was found to significantly influence the reaction rate. With SbF$_6^-$, which is less coordinating than NTf$_2^-$, oxidative addition of PhI occurs spontaneously and within the time of mixing at room temperature, the same Au(III) complex **2** is obtained as sole product. This result suggests that the first step of the oxidative addition process is the displacement of the counter-anion by iodobenzene to form a π- or I-adduct (see mechanistic discussions below). The formation of **2** shows that oxidative addition of PhI to gold is not limited to diphosphino-carborane (DPCb) complexes[37]. Besides generalizing the transformation to simpler and readily available Au(I) species, the Me-Dalphos complex **1** also displays enhanced reactivity. This is particularly striking for the oxidative addition of biphenylene (Fig. 3)[35, 38]. As previously reported, it is possible with DPCb-gold complexes but requires 5 h at 120 °C[37]. In comparison, oxidative addition of the C–C bond is complete and quantitative within 2 h at room temperature (RT) with **1**/AgSbF$_6$.

**Scope and mechanism of the reaction.** Oxidative additions of a series of aryl iodides were then tested (Fig. 4), and here also the reactivity of the (P,N) complex **1**/AgSbF$_6$ surpassed that of (DPCb) gold complexes[37]. Instantaneous reactions were observed with *para*-substituted iodobenzenes featuring electron-donating groups as well as electron-withdrawing groups. *Ortho*-substituted substrates such as iodonaphthalene and *ortho*-methyl iodobenzene are also efficiently activated, although complete conversion requires 1 h in these cases. In addition, the scope of aryl iodides was extended to electron-poor and sterically hindered substrates, such as iodopentafluorobenzene. With **1**/AgSbF$_6$, oxidative addition is complete and quantitative in 3.5 h at RT,

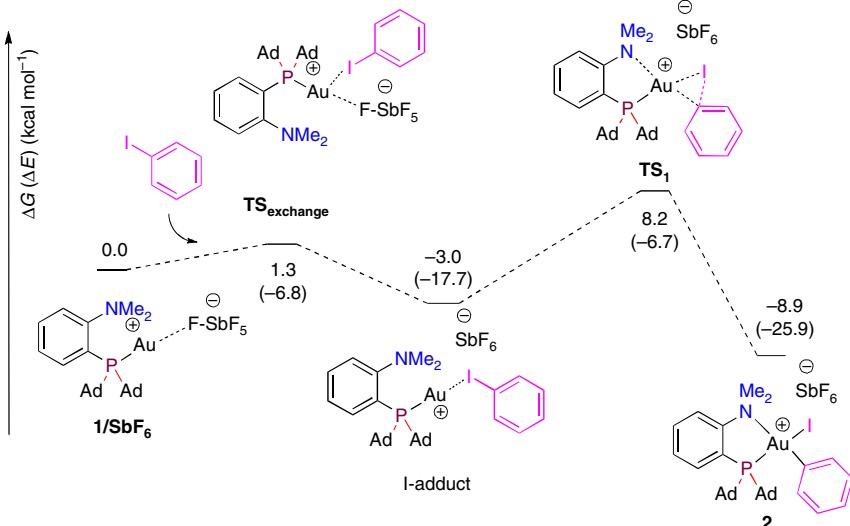

**Fig. 5** Computed energy profile for the oxidative addition of PhI to gold. Energy profile computed at the B97D(SDM-DCM)/SDD+f(Au), SDD(I,Sb),6-31G** (other atoms) level of theory taking into account the $SbF_6^-$ counter-anion and dichloromethane effects with Universal Solvation Model based on solute electron density (SMD model). Electronic energy ($\Delta E$) including ZPE correction into brackets and Gibbs free energy ($\Delta G$) in kcal mol$^{-1}$

whereas no reaction occurred with (DPCb) gold complexes[37]. Iodoheteroarenes such as 2-iodopyridine are also suitable substrates. Competitive nitrogen coordination slows down the process, but quantitative reaction is achieved over 3 days. Remarkably, oxidative addition of aryl bromides was also demonstrated with gold using 1/AgSbF$_6$. The reaction works well with both electron-rich and electron-poor substrates (as substantiated by the *para*-acetyl and *ortho*-methoxybromobenzenes) and the corresponding Au(III) aryl complexes are obtained in good to high yields (60–93%).

The mechanism of the reaction and the key role of the (P,N) ligand on the oxidative addition process were examined computationally at the B97D(SDM-DCM)/SDD+f(Au), SDD(I,Sb),6-31G** (other atoms) level of theory with the real complex 1/SbF$_6$ (Fig. 5) taking into account the counter-anion (SbF$_6^-$) and solvent effects (fully optimized structures in dichloromethane). As a first step of the reaction, the displacement of the weakly coordinating counter-anion SbF$_6^-$ by iodobenzene proceeds with a very low activation barrier ($\Delta G^{\neq}$=1.3 kcal mol$^{-1}$) to form a Au(I)–PhI adduct (I-adduct), which is slightly downhill in energy ($\Delta G$=–3 kcal mol$^{-1}$). The oxidative addition step then proceeds with an activation barrier $\Delta G^{\neq}$ of 11.2 kcal mol$^{-1}$ via the 3-center transition state **TS$_1$**. It is favored thermodynamically ($\Delta G$=–8.9 kcal mol$^{-1}$) and leads to the tetracoordinate Au(III) aryl complex 2 we obtained experimentally (the optimized structure matches very well that determined crystallographically). In comparison (Supplementary Fig. 49), NTf$_2^-$ binds more strongly to gold ($\Delta G_{1-NTf2\rightarrow PhI}$=11.3 kcal mol$^{-1}$). Its displacement by PhI requires an activation barrier ($\Delta G^{\neq}$=18.9 kcal mol$^{-1}$) which is significantly higher than that of the oxidative addition step ($\Delta G^{\neq}$=13.8 kcal mol$^{-1}$). This explains the strong counter-anion effect, the rate of the reaction being significantly slower with NTf$_2^-$ than SbF$_6^-$.

The formation of the other Au(III) diastereomer with the phenyl ring in *cis* position to nitrogen was also investigated theoretically (Supplementary Fig. 50). The reaction is less favored thermodynamically and requires a higher activation barrier. To further analyze the impact of the adjacent nitrogen atom, the reaction profile for oxidative addition of iodobenzene to the related gold complex devoid of NMe$_2$ substituent was then computed (Supplementary Fig. 51). The transformation is still

kinetically feasible, but the corresponding activation barrier is about twice as large. Moreover, the reaction is not favored thermodynamically ($\Delta G$=11.2 kcal mol$^{-1}$) due to the formation of a high-energy 3-coordinate Au(III) species. This comparison highlights the critical role of the hemilabile (P,N) ligand that lowers the activation barrier for oxidative addition and thermodynamically stabilizes the resulting Au(III) complex by coordination of the nitrogen atom.

**Development of catalytic arylation with aryl halides**. Having substantiated that oxidative addition of aryl halides to Au(I) occurs readily with 1/AgSbF$_6$, we then sought to go beyond this individual elementary step and showcase a Au(I)/Au(III) redox cycle enabling C–C cross-coupling. Remarkable gold-catalyzed oxidative coupling of arenes (and heteroarenes) with aryl silanes[12–16], aryl boronates[17, 18] and arenes[19, 20] have been recently reported. It is worth noting that these cross-coupling reactions generally proceed under milder conditions and with different selectivities than palladium-catalyzed processes. The site selectivity of the Csp$^2$–H activation step is imparted by highly electrophilic Au(III) intermediates (which react via an S$_E$Ar pathway) and no *ortho*-directing groups are employed. However, the gold-catalyzed transformations require an external oxidant, typically an I(III) derivative, to achieve the Au(I)/Au(III) redox cycle, and high selectivity in cross-coupling (versus homo-coupling) requires electronically different coupling partners (one electron-rich and one electron-poor aryl ring). Nonetheless, these pioneering contributions clearly show that Au(III) holds great potential for Csp$^2$–H bond arylation processes. It was thus very appealing to investigate the reactivity of the Au(III) aryl complexes generated by oxidative addition of aryl halide for direct arylation processes.

As a prototype example, we evaluated the coupling between iodobenzene and 1,3,5-trimethoxybenzene (TMB) with 1/AgSbF$_6$. The transformation was first assessed under stoichiometric conditions, by reacting the gold(III) aryl complex 2 intermediate derived from oxidative addition of PhI, with 1 equiv. of TMB. Under these conditions, the desired cross-coupling product forms very slowly, as deduced from gas chromatography–mass spectrometry (GC-MS) analysis. However, addition of 1 equiv. of

**Fig. 6** Cross-coupling of iodobenzene and 1,3,5-trimethoxybenzene with 1/AgSbF₆. **a** Sequential addition of reagents. **b** One-step conditions

AgSbF₆—to abstract the iodine from **2** and generate a more electrophilic Au(III) species—afforded the biaryl product in 90% isolated yield after 12 h at room temperature (Fig. 6a). Encouragingly for the intended catalytic development, the coupling reaction proceeded equally well sequentially and in one step (Fig. 6b). At this stage, different solvents were screened to try to improve the efficiency and the rate of the cross-coupling reaction (Supplementary Table 1). Working in dichloromethane with a small amount of MeOH (50:1) gave the best result. Under these conditions, the biaryl product was obtained in 97% yield within 5 h at RT. The next step was to translate the cross-coupling from stoichiometric to catalytic conditions. The feasibility of the catalytic transformations was first demonstrated using 20 mol% of the gold complex **1** and 1.2 equiv. of AgSbF₆. The coupling product was obtained in 82% yield after 14 h at 75 °C. Based on this promising result, we surveyed the influence of different parameters—the catalytic loading, the ratio of the coupling partners, the temperature and reaction time, the halide scavenger, the solvent and the presence of a base—and came up with the following optimal conditions: 5 mol% of **1**, 1 equiv. of TMB, 1 equiv. of AgSbF₆, 75 °C, 2 h, dichlorobenzene (DCB)/MeOH (50:1) and K₃PO₄ (1 equiv.) (Supplementary Tables 2–4). Under these conditions, the coupling product was obtained in 91% yield. Note that it is possible to further decrease the catalytic loading. The biaryl product was also obtained in 90% yield using 1 mol% of the gold complex **1** after keeping overnight at 75 °C.

**Scope of the reaction and mechanistic considerations**. Having set good conditions for the catalytic cross-coupling of iodobenzene and TMB with complex **1**, we screened the scope of aryl halides (Table 1). The arylation of TMB proceeded well with diverse iodobenzenes (bearing electron-poor or electron-rich *para*-substituents) as well as with 8-iodonaphthalene. The corresponding biaryl products were obtained in good to excellent yields under mild conditions (75 °C, 2 h). The ability of complex **1** to undergo oxidative addition of aryl bromides could also be transposed to catalytic cross-coupling. Using 10 mol% of gold and 5 equiv. of Ar–Br (to keep mild conditions and reasonably short reaction times), the biaryl products were also obtained in good yields. Particularly noteworthy is the coupling of the *para*- and *ortho*-OMe substituted Ar–X substrates with TMB, which afford biaryl products with electron-rich substituents at each aromatic ring. Thus, the scope of biaryl products accessible by this Ar–X/Ar'–H coupling strategy is complementary to that of the oxidative coupling reactions recently developed by Lloyd-Jones, Russell and colleagues[12–15], Nevado and colleagues[17] and Larrosa and

colleagues[19, 20] which proceed well when one of the coupling partner is electron deprived.

Given the afore-described stoichiometric reactions, these arylation reactions most probably operate via 2-electron redox catalytic cycles. To further support this mechanistic scenario under catalytic conditions, some additional control experiments were carried out under day light or in the dark, with or without Hg(0) (as a trap for heterogeneous metal particles, Supplementary Fig. 1) and with or without galvinoxyl (as a radical trap, Supplementary Fig. 2). In neither case, the kinetic and efficiency of the coupling reaction were affected, making light-mediated, heterogeneous and radical paths very unlikely.

Finally, some tests were performed to extend the gold-catalyzed coupling to the arylation of heteroarenes. Encouragingly, 1-phenylpyrrole and iodobenzene were successfully coupled using 5 mol% of **1**/AgSbF₆ under our standard mild conditions (75 °C, 2 h, Fig. 7). Without further optimization, the reaction gave a 61% yield with high β-selectivity (β:α=9/1). Transition metal-mediated arylations of pyrroles usually require higher reaction temperatures and proceed preferentially at the α-position[52]. β-Arylation of pyrroles is very much looked after but comparatively rare[13, 53]. Although preliminary, these results indicate that our gold-catalyzed pathway for direct arylation has some generality and is certainly worthwhile to be explored further to complement known arylation methodologies.

In summary, we have demonstrated that the bidentate (P,N) ligand Me-Dalphos is very efficient in promoting oxidative addition to gold. The reaction proceeds rapidly under mild conditions with a large scope of aryl iodides and bromides. Density functional theory calculations have shed light on the reaction mechanism and on the key role of the nitrogen atom. The ensuing Au(III) complexes smoothly react with trimethoxybenzene to form the corresponding biaryl products. Such a 2e-redox sequence was transposed to catalytic conditions and implemented in a Au(I)/Au(III) cross-coupling reaction affording biaryls. From practical and synthetic viewpoints, this transformation is complementary to the recently developed gold-catalyzed oxidative arylations. It involves aryl halides as electrophiles[54], it tolerates electron-donating as well as electron-withdrawing substituents and it does not require an external oxidant to generate the key Au(III) catalytic intermediate[55, 56]. Due to their ability to promote oxidative addition to gold and stabilize Au(III) species, hemilabile ligands such as Me-Dalphos hold great potential in gold-catalyzed arylation reactions. Future work will seek to generalize this ligand design principle to other classes of bidentate ligands, to extend the scope of electron-rich (hetero)arenes and to develop further gold-catalyzed cross-coupling reactions with aryl halides.

**Table 1 Gold-catalyzed arylation of 1,3,5-trimethoxybenzene with aryl halides[a]**

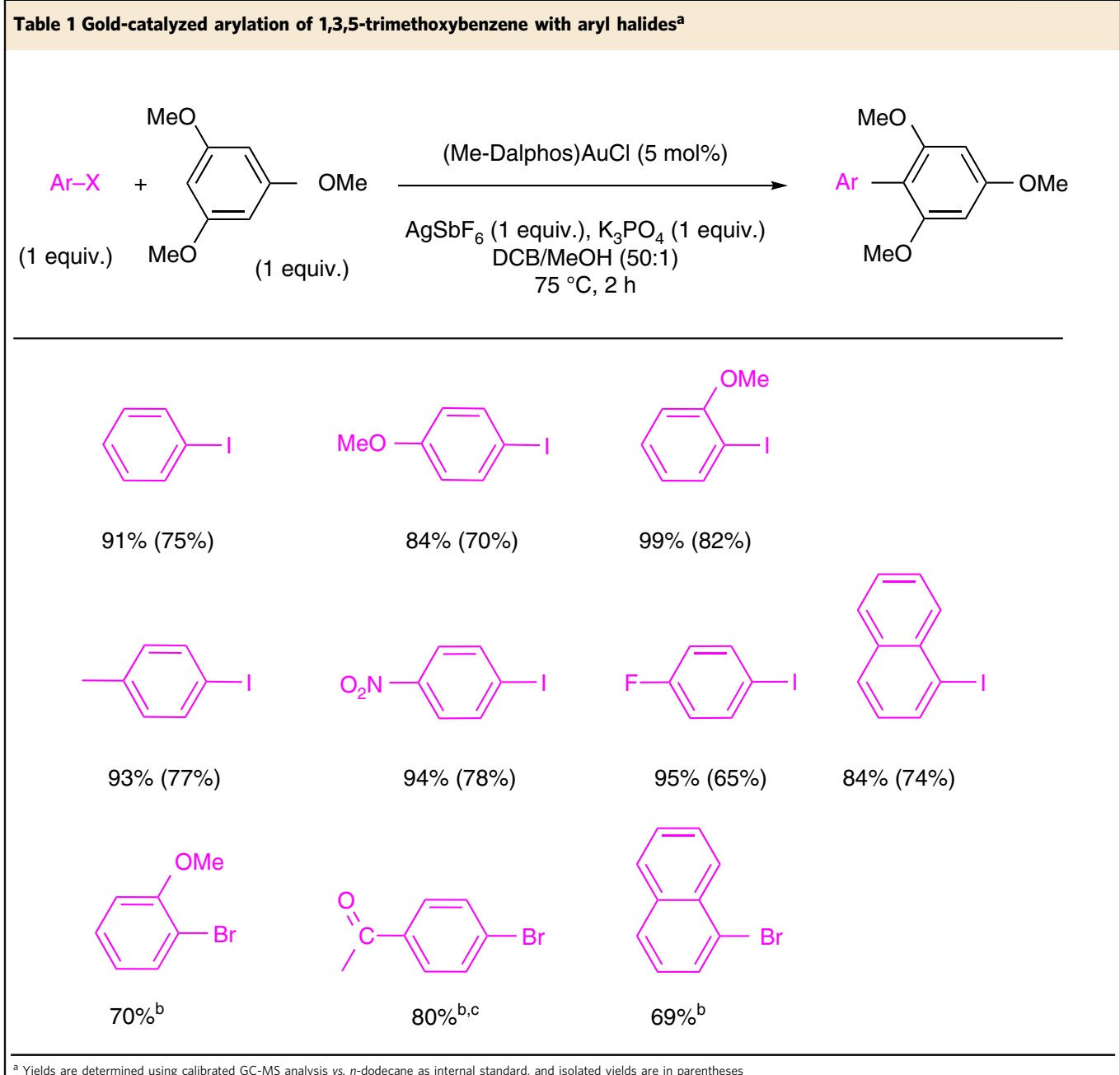

[a] Yields are determined using calibrated GC-MS analysis *vs. n*-dodecane as internal standard, and isolated yields are in parentheses
[b] [Au] 10 mol% and 5 equiv. of ArBr
[c] Reaction time 12 h

## Methods

**General information**. Unless otherwise stated, all reactions and manipulations were carried out under an atmosphere of dry argon using standard Schlenk techniques or in a glovebox under an inert atmosphere. Dry, oxygen-free solvents were employed. All starting materials were purchased from Aldrich and used as received unless otherwise stated.

**General procedure for the oxidative addition of aryl iodides**. In a glovebox, a screw-cap NMR tube was charged with silver hexafluoroantimonate (8.0 mg, 0.023 mmol) in dichloromethane-d2 (0.3 ml). Complex **1** (15 mg, 0.023 mmol) was transferred into a small glass vial and dissolved in dichloromethane-d2 (0.3 ml). The aryl iodide (0.115 mmol) was added to the solution of **1**. The prepared solution was loaded into a plastic syringe equipped with stainless steel needle. The syringe was closed by blocking the needle with a septum. Outside the glovebox, the NMR tube was cooled down to –80 °C (Ethanol/N₂ cold bath). At this temperature, the solution of complex **1** and aryl iodide was added. The tube was gently shaken and allowed to warm to RT. The reaction was left to proceed until completion as monitored by ³¹P (¹H) NMR. The formation of the Au(III) complex was confirmed

by ¹H and ³¹P NMR spectroscopy and high-resolution mass spectrometry (electrospray ionization, positive mode).

**General procedure for direct arylation of trimethoxybenzene with aryl iodides**. In a glovebox, a flame-dried Schlenk equipped with a magnetic stirrer bar was charged with silver hexafluoroantimonate (144 mg, 0.42 mmol) and potassium phosphate tribasic (85 mg, 0.40 mmol) in DCB (2.0 ml). Complex **1** (13 mg, 0.02 mmol) was transferred into a small glass vial and dissolved in DCB (2.0 ml). Aryl iodide (0.4 mmol, 1 equiv.), TMB (67.0 mg, 0.4 mmol) and methanol (80 µL) were added to the gold complex solution. This solution was loaded into a plastic syringe equipped with stainless steel needle. The syringe was closed by blocking the needle with a septum. Outside the glovebox, the Schlenk was cooled down to –10 °C (Ethanol/N₂ cold bath). At this temperature, the solution of complex **1**, aryl iodide and TMB was added. The reaction mixture was then stirred at 75 °C. The yields were determined by GC-MS using *n*-dodecane as an internal standard. After complete conversion, silver salts were filtrated, and the solvent evaporated. The sample was purified by column chromatography (pentane/ethyl acetate). The fractions containing the biaryl product were then concentrated in vacuo to yield the pure product.

**Fig. 7** Gold-catalyzed arylation of 1-phenylpyrrole with iodobenzene. Reaction carried out using the (Me-Dalphos) AuCl complex **1** (5 mol%) in the presence of AgSbF$_6$ and K$_3$PO$_4$

**Data availability**. The authors declare that the data supporting the findings of this study are available within the article and the accompanying Supplementary Information Files, which are both free of charge to access. For detailed experimental procedures, spectroscopic and physical data of compounds, see Supplementary Methods, Supplementary Figs. 1, 2 and Supplementary Tables 1–5. For NMR spectra of the compounds in this article see Supplementary Figs. 3–47. The CCDC 1531070 (**3**), 1531071 (**6**), 1531072 (**7**), 1531073 (**9**) and 1531074 (**12**) contain the supplementary crystallographic data for this paper (Supplementary Fig. 48 and Supplementary Tables 5, 6). These data can be obtained free of charge from The Cambridge Crystallographic Data Centre via www.ccdc.cam.ac.uk/data_request/cif. For computational details and Z-matrices see Supplementary Methods, Supplementary Figs. 49–52 and Supplementary Tables 7–10. All these data are available from the corresponding authors upon reasonable request.

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

## Acknowledgements

Financial support from the Centre National de la Recherche Scientifique, the Université de Toulouse and the Agence Nationale de la Recherche (ANR-JCJC-2012-POGO) is gratefully acknowledged. We thank Umicore AG & Co for a generous gift of gold precursors. UPPA, MCIA (Mésocentre de Calcul Intensif Aquitain) and IDRIS under Allocation 2016 (i2016080045) made by Grand Equipement National de Calcul Intensif (GENCI) are acknowledged for computational facilities. L.E. thanks the Xunta de Galicia for financial support through the I2C program.

## Author contributions

A.A. and D.B. conceived and designed the study; A.Z. performed the experiments; L.E. carried out the calculations; A.Z., A.A. and D.B. analyzed the experimental data; L.E. and K.M. analyzed the computational data; and S.M.-L. collected and refined the X-ray diffraction data. All the authors contributed to scientific discussion. A.A. and D.B. wrote the manuscript.

## Additional information

**Competing interests:** The authors declare no competing financial interests.

