## [Peer Review File · Nature Communications]

Reviewers' comments:

Reviewer #1 (Remarks to the Author):

This work by Bourissou and co-workers describes the smart use of a P,N hemilabile ligand to favor Au(I)/Au(III) redox processes and its direct application to catalytic cross-coupling reactions. Although there are a number of examples on gold-mediated cross-coupling processes, the present work constitutes a highly important addition to the field by defining a clear and simple strategy to facilitate oxidative addition of aryl halides over Au(I) centers, certainly one of the bottlenecks on gold-catalyzed cross-coupling chemistry. The use of the hemilabile ligand both kinetically favors oxidative addition reactions and also stabilizes the resulting three-coordinated Au(III) compounds by N-ligation, the latter usually too reactive to design efficient Au(I)/Au(III) catalytic redox cycles. The authors describe more than ten aryl iodides and bromides that oxidatively add to Au(I) in excellent yields and with remarkable functional group tolerance. This is indeed an additional virtue of the system since, contrary to prior examples, it permits accessing biaryl products with two electron-rich aromatic rings. The authors provide three well-characterized catalytic examples. Although the substrate scope regarding the C-H activated aryl ring is only limited to trimethoxybenzene and 1-phenyl-pyrrole, this does not detract from the importance of the overall results. The manuscript is very-well written and pleasant to read, the characterization of the new compounds is exhaustive and I appreciate that NMR spectra are included in the Supporting Information. DFT studies are very informative about the core subject, namely oxidative addition over Au(I) centers. Considering all the above and taking into account that gold catalysis and, in particular, cross-coupling reactions mediated by gold are at the forefront of current challenges in catalysis, I give my strongest support for the acceptance of this article in Nature Communications, once the following very minor aspects are considered by the authors.

- The last sentence of the introduction suggests '...the development of the first gold-catalyzed C-C cross-coupling from aryl halides'. In this sense I would also consider Sonogashira reaction in *Angew. Chem. Int. Ed.* 2007, 46, 1536 and *Eur. J. Org. Chem.* 2008, 5946 by Corma and Wang, respectively. Although the present work offer a much more detailed mechanistic understanding on the process, those previous contributions may also be considered as C-C cross-coupling reactions from aryl halides.
- I am unaware of the preferred number of Figures and Schemes for an article of this length in *Nat. Commun.*, however, if possible, I would recommend to include a Scheme in page 5 to visualize the rather interesting oxidative addition of biphenylene. An additional Scheme on page 10 for the coupling of iodobenzene and pyrrole would also be clarifying.
- Page 6. Substitute 'I-adduct' by 'iodobenzene adduct' or define 'I-adduct' in Fig. 2 (it only shows up in the SI). Also, it would be helpful to explicitly point to Fig. S49 rather than just ref 49 after 11.3 kcal.mol⁻¹.
- Page 6 and 7. It seems that the DG values of -8.9 and 12.7 refer to the same step, however, while the former corresponds to the transit from Sbf₆ adduct to compound 2, the latter refers to its formation from I-adduct. To be consistent both values should refer to the same process.
- As a curiosity, I wonder if the hemilabile amine group participates as well in the C-H activation of trimethoxybenzene by abstracting the proton. I will look forward to see an overall mechanistic picture and further applications of the concept in future contributions.

Reviewer #2 (Remarks to the Author):

The authors carry out a combined computational-experimental study on a very interesting topic, which would be of interest to the scientific community.

A good level of theory is used to integrate the experimental outcome with computations and thus examine in more detail the possible mechanistic hypothesis. The overall organization of the paper is satisfactory.

I would recommend publication after revising the following points:

1) It is not clear to me if the reported ΔG 's include the solvent effect or not, that is if they were obtained by frequency calculations performed in gas phase or in solvent?

2) How do the authors explain the large differences between the electronic and Gibbs energies profiles, in terms of relative ΔE and ΔG ? The discrepancies are quite relevant and deserve comments by the authors.

3) The displacement of the SbF_6^- counterion by iodobenzene appears to be barrierless. This seems quite strange: although the anion is weakly coordinating the gold center, there are definitely electronic and also steric effects (the second gold ligand is quite cumbersome and, given the strong gold preference for a linear coordination, the direction of approach of the incoming ligand is restrained), which should quite likely give rise to an activation barrier.

4) The authors explain the experimental slower reaction rate observed with the bistriflimide focusing on the first reaction step. However, with any counteranion, the rate-determining step seems to be the second one (Figure S49), that is the oxidative addition itself. Thus, the reason of the different experimental outcome should be searched, above all, in an analysis of the two TS1 transition states. That is: the authors should explain why the two TS1 transition states with the two counter-anions have such a different stability with respect to the corresponding reactants. It would also be helpful to discuss also the differences between the I-adducts

5) page 3: the authors refer to "the high redox potential of the Au(I)/Au(III) couple compared to that of Pd(0)/Pd(II)" as the main reason of the Au(I) complexes difficulty to promote oxidative addition. It could be useful to mention other references and not only palladium for making this statement

6) it would be helpful if the authors could explain why the reaction does not occur with $[(\text{DPCb})\text{Au}]^+$ as stated on page 5

7) Do the author identify any trend/pattern when carrying out the reaction with the various species displayed in Scheme2?

8) The authors points out that it is key for the success of their procedure the presence of the coordinating N close to the gold center, which stabilizes the Au(III) adduct. Although Au(I) strongly prefers a 2-ligand coordination, it is also true that it can accept electronic density also in this oxidation state: do the authors observe any coordination by the NMe2 group before the oxidative addition?

Minor points:

1) page 2: there are 3 dots at the end of the first paragraph: I assume it is a typo

2) page 2: the last sentence of the second paragraph seems confusing to me and should be rephrased

3) caption of Scheme 1: something seems to be missing when reads: "Au-P [0]"

Best Regards

Gian Pietro Miscione

Reviewer #3 (Remarks to the Author):

This manuscript describes results that expand upon previous work by this group on the oxidative addition of aryl iodides to gold(I) to form Au(III) complexes. What is new here is the simpler ligand design, a broader reaction scope and the demonstration that the resulting aryl-Au(III) complexes react with 1,3,5-trimethylbenzene both under stoichiometric and catalytic conditions. The observation of a room temperature oxidative addition of biphenylene to Au(I) is also interesting (ref. 31 should also have been cited in p. 5 within this context).

However, although biaryls can be obtained from aryl iodides, this is not a "cross-coupling" as stated in the title of the manuscript and elsewhere, but rather an intermolecular arylation reaction. Cross-couplings are reactions between alkyl, alkenyl or aryl halides R'-X and organometallic reagents to form R-R'. Although this distinction might be considered by some as formal, formalities are important not to mislead the potential readers of this work.

The authors state in the introduction that "Monocoordinate gold(I) cationic complexes [LAu]⁺ (L = R₃P or NHC) have also been shown to undergo oxidative addition (refs, 34, 41, 42)". This statement is not correct (at least) for reactions proceeding in solution. In ref. 41, the group of Toste demonstrated that NHCAuCl reacts with biphenylene by oxidative addition in the presence of AgSbF₆ by forming an intermediate aryl complex, not via a "monocoordinate gold(I) cationic complexes [LAu]⁺". Furthermore, in ref. 41 (see Figure 3), a "naked" [LAu]⁺ complex does not exist in the calculated reaction coordinate the oxidative addition. Finally, the evidence provided by the gas-phase study reported in ref. 42, actually indicates that if a naked [LAu]⁺ species would be generated in the gas-phase, its reaction with PhI would actually give first a two-coordinated [R₃PAu(PhI)]⁺ species.

Calculations are reported to support the relatively facile oxidative addition. Although those performed from the triflimide complex (Figure S49) are reasonable, the calculations shown in the body of the manuscript start from LAu-SbF₆ complex, whose actual involvement as real species under the reaction conditions is rather unlikely. More probably, the intermediate [L-Au-IPh]⁺ complex is formed by an associative chloride substitution from L-AuCl assisted by Ag⁺. Although the involvement of an arene-gold(I) complex as intermediate species is considered in the SI (Figure S51), this is proposed to arise from totally unrealistic monocoordinated LAu⁺. In addition, although experts in the field might understand that Au(I) is actually bound to the halogen in LAu-SbF₆ and other complexes shown in Figure 1 with SbF₆ or GaCl₄ as the ligands, the depiction chosen by these authors for the bonding is, at least, very misleading.

Supporting information:

In general, yields given for the described new complexes are based on NMR and not on the isolated pure compounds. Furthermore, elemental analyses were not reported for any of the gold(III) complexes, which is below the standards required for characterization of organometallic compounds. Three of the complexes resulting from oxidative addition of aryl bromides reported in p. S12 were only characterized by ³¹P NMR and HRMS.

The same lack of experimental rigor applies for the formation of the biaryls. In all cases, yields of 2,4,6-trimethoxy-1,1'-biphenyl were determined by GC. For the other biphenyls described in pp. S18-19, it is not clear if the yields correspond to isolated materials. All the melting points for these biaryls are missing. Biphenyl 22 is reported as a new compound, although it has been reported: *Tetrahedron Lett.* 2014, 55, 3184; 2016, 57, 4235. Melting point and HRMS (or elemental analysis) are missing for this compound.

For the two diphenyl-1H-pyrroles, which were obtained as a mixture, only ¹³C NMR spectra are reported.

The ³¹P NMR spectra of 2, 1, 13, 14 shows another minor signal. The ¹H NMR spectra of 3, 4, 5, 6, 8, 9, 10, 11, 12 show a large amount of ArI and/or other impurities.

Only two melting points are actually reported in this work (for complexes 13, 14). The statement that melting points are uncorrected is not correct. See: *Tiers, G. J. Chem. Educ.* 1990, 67, 258.

Minor corrections: 1-phenyl-pyrrole should be 1-phenylpyrrole

The weakest point of this study is that scope of the C-C bond formation is essentially limited to

1,3,5-trimethylbenzene as the electron-rich partner. In addition, the quality of the experimental data is well below the standards required for publication of organometallic and/or organics chemistry work in specialized journals such as *Organometallics*, *Dalton*, or *Org. Lett.*

As indicated above, this work contains sufficient elements of novelty to be published. However, considering all the precedents reported by this and other groups, the rather preliminary nature of the calculations, and the poor quality of the characterization data, publication of this work in *Nature Communications* is not recommended.

Point-by-point response to the referee's comments

Reviewer #1

This work by Bourissou and co-workers describes the smart use of a P,N hemilabile ligand to favor Au(I)/Au(III) redox processes and its direct application to catalytic cross-coupling reactions. Although there are a number of examples on gold-mediated cross-coupling processes, the present work constitutes a highly important addition to the field by defining a clear and simple strategy to facilitate oxidative addition of aryl halides over Au(I) centers, certainly one of the bottlenecks on gold-catalyzed cross-coupling chemistry. The use of the hemilabile ligand both kinetically favors oxidative addition reactions and also stabilizes the resulting three-coordinated Au(III) compounds by N-ligation, the latter usually too reactive to design efficient Au(I)/Au(III) catalytic redox cycles. The authors describe more than ten aryl iodides and bromides that oxidatively add to Au(I) in excellent yields and with remarkable functional group tolerance. This is indeed an additional virtue of the system since, contrary to prior examples, it permits accessing biaryl products with two electron-rich aromatic rings. The authors provide three well-characterized catalytic examples. Although the substrate scope regarding the C-H activated aryl ring is only limited to trimethoxybenzene and 1-phenyl-pyrrole, this does not detract from the importance of the overall results. The manuscript is very-well written and pleasant to read, the characterization of the new compounds is exhaustive and I appreciate that NMR spectra are included in the Supporting Information. DFT studies are very informative about the core subject, namely oxidative addition over Au(I) centers. Considering all the above and taking into account that gold catalysis and, in particular, cross-coupling reactions mediated by gold are at the forefront of current challenges in catalysis, I give my strongest support for the acceptance of this article in Nature Communications, once the following very minor aspects are considered by the authors.

1. The last sentence of the introduction suggests ‘...the development of the first gold-catalyzed C-C cross-coupling from aryl halides’. In this sense I would also consider Sonogashira reaction in *Angew. Chem. Int. Ed.* 2007, 46, 1536 and *Eur. J. Org. Chem.* 2008, 5946 by Corma and Wang, respectively. Although the present work offer a much more detailed mechanistic understanding on the process, those previous contributions may also be considered as C-C cross-coupling reactions from aryl halides.

We fully agree that the works of Corma (2007) and Wang (2008) on gold-catalyzed Sonogashira reaction also stand as examples of C-C cross-coupling reactions from aryl halides with gold. And the very recent contribution of Hierso (Chem. Asian J. 2017, 12, 459-464; Suzuki-Miyaura reaction with digold complexes) can also be added to the list. But in none of these reactions, the nature of the active species is known and the 2007 report of Corma et al has actually been the matter of a lively debate in the scientific community about the feasibility of oxidative addition of Ar-I to gold. Follow-up studies by different groups, including Corma's (Chem. Commun. 2011, 47, 1446) have pointed out the role of gold clusters/nanoparticles (the activation of Ar-I does not proceed by oxidative addition at a single metal center, but rather by cooperative addition to neighboring gold atoms). This being said, the last sentence has been reformulated as follows: "The potential of (P,N) gold(I) complexes in Au(I)/Au(III) catalysis is also demonstrated by the development of the first C-C cross-coupling reactions involving aryl halides and catalyzed by well-defined mononuclear gold complexes." and the papers by Corma, Wang and Hierso have been cited as references 49-51.

2. I am unaware of the preferred number of Figures and Schemes for an article of this length in Nat. Commun., however, if possible, I would recommend to include a Scheme in page 5 to visualize the rather interesting oxidative addition of biphenylene. An additional Scheme on page 10 for the coupling of iodobenzene and pyrrole would also be clarifying.

We have included two additional schemes following the suggestion of the reviewer

- Scheme 2. Oxidative addition of biphenylene to gold

- Scheme 5. Cross-coupling of iodobenzene and 1-phenylpyrrole catalyzed by the (Me-Dalphos)AuCl complex **1** in the presence of AgSbF₆.

3. Page 6. Substitute 'I-adduct' by 'iodobenzene adduct' or define 'I-adduct' in Fig. 2 (it only shows up in the SI).

The term I-adduct has been defined in Figure 2.

4. Also, it would be helpful to explicitly point to Fig. S49 rather than just ref 49 after 11.3 kcal.mol⁻¹.

The reader is now referred to the Figure S49 of the Supporting Information (comparison of the Au/anion binding energies and reactions profiles of PhI oxidative addition).

5. Page 6 and 7. It seems that the DG values of -8.9 and 12.7 refer to the same step, however, while the former corresponds to the transit from SbF₆ adduct to compound 2, the latter refers to its formation from I-adduct. To be consistent both values should refer to the same process.

*We fully agree with the reviewer. The idea here was to analyze the impact of the adjacent NMe₂ group on the oxidative addition process. For that purpose, we carried out calculations with complex **1**_H* devoid of NMe₂ group. To save calculation time, the process was initially computed with the model cationic gold complex in the gas phase without taking into account the counter anion. In that case it was more consistent to use the I-adduct as reference rather than the unstable naked cationic gold complex. However, following the recommendations of the reviewer, we have carried out new calculations for the oxidative addition of PhI to **1**_H/SbF₆ at the same level of theory as for **1**/SbF₆ taking into account the counter-anion and solvent effect (DCM) so that the two reaction profiles (with/without NMe₂) are fully consistent. Figure S51 of the Supplementary Information displays the new reaction profile and the ΔG values for the exact same step of the two profiles are now indicated in the main text (ΔG = -8.9 kcal.mol⁻¹ for **1**/SbF₆ vs ΔG = 11.2 kcal.mol⁻¹ for **1**_H/SbF₆).*

6. As a curiosity, I wonder if the hemilabile amine group participates as well in the C-H activation of trimethoxybenzene by abstracting the proton. I will look forward to see an overall mechanistic picture and further applications of the concept in future contributions.

Indeed, it is possible to envision some participation of the NMe₂ group in the C-H activation of TMB, but the strong N-Au(III) interaction involved in the oxidative addition product makes this pathway unlikely. It can also be noticed that even in the stoichiometric reactions carried out without external base, we did not observe the protonation of the amine.

Reviewer #2

The authors carry out a combined computational-experimental study on a very interesting topic, which would be of interest to the scientific community. A good level of theory is used to integrate the experimental outcome with computations and thus examine in more detail the possible mechanistic hypothesis. The overall organization of the paper is satisfactory. I would recommend publication after revising the following points:

1. It is not clear to me if the reported ΔG 's include the solvent effect or not, that is if they were obtained by frequency calculations performed in gas phase or in solvent?

*Unless otherwise stated, the energy profiles were obtained by frequency calculations performed in solvent. We have clarified this point in the main text as follows: "The mechanism of the reaction and the key role of the (P,N) ligand on the oxidative addition process were examined computationally at the B97D(SDM-DCM)/SDD+f(Au), SDD(I,Sb),6-31G**(other atoms) level of theory with the real complex **1** (Figure 2) taking into account the counter-anion and solvent effects (fully-optimized structures in dichloromethane)."*

*This has also been precised in the computational details provided in the Supporting Information as follows (page S50): "Full optimizations were carried out for all stationary points, minima and transition state structures, involved in the reaction process: oxidative addition of PhI or PhBr to i) real complex **1** at the B97D(SDM-DCM)/SDD+f(Au), SDD(I),6-31+G**(other atoms) level of theory by taking into account the counter-anion [SbF₆], **1-SbF₆**, and [NTf₂], **1-NTf₂** and solvent effects by means of the solvation model SMD¹⁴ for dichloromethane (DCM, results reported in Figure 2, Figures S49 and S52 in ESI); ii) to the model compound **1H-SbF₆** at the B97D(SDM-DCM)/SDD+f(Au), SDD(I),6-31+G**(other atoms) level of theory taking into account solvent effect (Figure S51 in ESI); iii) to cationic species **1*** in gas phase at the B97D/SDD+f(Au), SDD(I,Sb), 6-31G**(other atoms) level of theory (Figures S50 in ESI). In each case, frequency calculations were undertaken at the same level of theory to confirm the nature of the stationary points, yielding one imaginary frequency for transition states (TS), corresponding to the expected process, and zero for minima."*

2. How do the authors explain the large differences between the electronic and Gibbs energies profiles, in terms of relative ΔE and ΔG ? The discrepancies are quite relevant and deserve comments by the authors.

*The difference between the electronic and Gibbs energies is strongly dependent on the molecularity of the associated process that contributes significantly to the entropic term. While unimolecular reactions have indeed a free energy value close to the enthalpy and electronic energy values, a significant difference is generally observed for multimolecular processes due to entropic effects. Indeed, entropic effects tend to overestimate the energetic costs of bimolecular processes, a difference between ΔE and ΔG values around 10 to 20 kcal·mol⁻¹ being classically observed (See for example L. A. Watson & O. Eisenstein, *J. Chem. Ed.* **2002**, 79(10), 1269). Thus, the 14 kcal·mol⁻¹ difference observed between the ΔE and ΔG values for the reaction of **1-SbF₆** with PhI is in the expected range for a bimolecular process and essentially comes from the translational entropy of the system.*

3. The displacement of the SbF₆⁻ counterion by iodobenzene appears to be barrierless. This seems quite strange: although the anion is weakly coordinating the gold center, there are definitely electronic

and also steric effects (the second gold ligand is quite cumbersome and, given the strong gold preference for a linear coordination, the direction of approach of the incoming ligand is restrained), which should quite likely give rise to an activation barrier.

*The reaction profile for the displacement of SbF_6^- by PhI at gold is very flat and all our attempts to localize a transition state for it long remained unsuccessful (in contrast with the displacement of NTf_2^- by PhI for which the TS could be relatively easily localized). Following the reviewer recommendations, we have further worked on it and mapped extensively the potential energy surface, initially imposing some constraints (on the Au–F and Au–I bond distances). This finally allowed us to localize a TS for the displacement of SbF_6^- by PhI. It is indeed very low in energy ($\Delta G^\ddagger = 1.3 \text{ kcal}\cdot\text{mol}^{-1}$) confirming that this is essentially a barrierless process, the rate-determining step of the reaction of PhI to **I-SbF₆** being the oxidative addition to gold. Similar additional calculations have been performed for the reaction of the model complex **I_H-SbF₆** (free of NMe₂ group) with PhI and for the reaction of **I-SbF₆** with PhBr in order to provide a consistent picture of all the considered reactions.*

*Figure 2 in the main text has been modified accordingly. Figure S49 (comparison of the energy profiles for the reactions of **I-SbF₆** and **I-NTf₂** with PhI), Figure S51 (comparison of the energy profiles for the reactions of **I-SbF₆** and **I_H-SbF₆** with PhI) and Figure S52 (energy profile for the reaction of **I-SbF₆** with PhBr) of the Supporting Information have been implemented similarly.*

4. The authors explain the experimental slower reaction rate observed with the bistriflimide focusing on the first reaction step. However, with any counteranion, the rate-determining step seems to be the second one (Figure S49), that is the oxidative addition itself. Thus, the reason of the different experimental outcome should be searched, above all, in an analysis of the two TS1 transition states. That is: the authors should explain why the two TS1 transition states with the two counter-anions have such a different stability with respect to the corresponding reactants. It would also be helpful to discuss also the differences between the I-adducts

*The fact we had not localized the TS for the displacement of SbF_6^- by PhI and the way we superposed the energy profiles for the reactions of **I-SbF₆** and **I-NTf₂** with PhI in the first version made the discussion of the counteranion effect quite confusing, we apologize for that. As stated above, the localization of the TS for the displacement of SbF_6^- by PhI now makes clear that this is essentially a barrierless process and that the rate-determining step of the reaction of PhI to **I-SbF₆** is indeed the oxidative addition to gold ($\Delta G^\ddagger = 11.2 \text{ kcal}\cdot\text{mol}^{-1}$ from I-adduct). The situation is different with the NTf_2^- counteranion. It is more strongly bound to gold and its displacement by PhI proceeds with a higher activation barrier ($\Delta G^\ddagger = 18.9 \text{ kcal}\cdot\text{mol}^{-1}$, Figure S49). This displacement becomes the rate-determining step in this case. Indeed, the barrier for the oxidative addition of PhI to **I-NTf₂** ($\Delta G^\ddagger = 13.8 \text{ kcal}\cdot\text{mol}^{-1}$ from the I-adduct again) is about 6 kcal/mol lower.*

*Please note also that the barriers for the oxidative addition of PhI to **I-NTf₂** and **I-SbF₆** are very similar, in line with the presence of only weak cation/anion interactions in the I-adducts and in the oxidative addition products. Consistently, the structures of the two transition states **TS₁** for the oxidative addition step are very similar for both counteranions.*

To clarify the impact of the counteranion displacement on the rate of the reaction, the text has been modified as follows: “As a first step of the reaction, the displacement of the weakly coordinating counter anion SbF_6^- by iodobenzene proceeds with a very low activation barrier ($\Delta G^\ddagger = 1.3 \text{ kcal}\cdot\text{mol}^{-1}$) to form a gold(I)–PhI adduct (I-adduct), which is slightly downhill in energy ($\Delta G = -3 \text{ kcal}\cdot\text{mol}^{-1}$)

¹). The oxidative addition step then proceeds with an activation barrier ΔG^\ddagger of 11.2 kcal.mol⁻¹ via the 3-center transition state **TS1**. It is favored thermodynamically ($\Delta G = -8.9$ kcal.mol⁻¹) and leads to the 4-coordinate gold(III) aryl complex **2** we obtained experimentally (the optimized structure matches very well that determined crystallographically). In comparison, NTf₂⁻ binds more strongly to gold ($\Delta G_{I-NTf_2 \rightarrow PhI} = 11.3$ kcal.mol⁻¹, see figure S49). Its displacement by PhI requires an activation barrier ($\Delta G^\ddagger = 18.9$ kcal.mol⁻¹, see figure S49) which is significantly higher than that of the oxidative addition step ($\Delta G^\ddagger = 13.8$ kcal.mol⁻¹).⁴⁹ This explains the strong counteranion effect, the rate of the reaction being significantly slower with NTf₂⁻ than SbF₆⁻.”

The way the energy profiles for the reactions of **1-SbF₆** and **1-NTf₂** with PhI are superposed in the Figure S49 in the Supporting Information has also been changed. To make the comparison easier, the I-adducts are now used as references and all energies are referenced to them. This highlights the similarity of the oxidative addition step. The very difference comes from the different binding strength of the two counteranions to gold in the initial displacement of SbF₆⁻ or NTf₂⁻ by PhI.

5. page 3: the authors refer to “the high redox potential of the Au(I)/Au(III) couple compared to that of Pd(0)/Pd(II)” as the main reason of the Au(I) complexes difficulty to promote oxidative addition. It could be useful to mention other references and not only palladium for making this statement

The comparison of the redox potentials has been extended to Pt(0)/Pt(II). The cited reference (ref 32) contains several other examples.

6. It would be helpful if the authors could explain why the reaction does not occur with [(DPCb)Au]⁺ as stated on page 5

This statement refers to the reaction between [(DPCb)Au]⁺ and iodopentafluorobenzene. It is difficult to provide a clear explanation at this stage. In our previous study on oxidative addition of aryl iodides to [(DPCb)Au]⁺ (ref 37), para-substituted iodobenzenes were screened and electron-withdrawing groups (EWG) were found to considerably slow down the reaction. Following this idea, we surmise that iodopentafluorobenzene (which is an electron-poor and sterically hindered substrate) reacts too slowly with [(DPCb)Au]⁺ to compete with decomposition of the cationic species [(DPCb)Au]⁺.

7. Do the author identify any trend/pattern when carrying out the reaction with the various species displayed in Scheme2?

*Yes, as presented in scheme 2, complex **1/SbF₆** reacts almost instantaneously with para-substituted iodobenzenes, but is sensitive to sterics. Longer reaction times are required for naphthalene and ortho-substituted substrates. The activation of 2-iodopyridine also shows the compatibility with N-containing substrates.*

The following sentences have been added to the discussion: “Ortho-substituted substrates such as iodonaphthalene and o-methyl iodobenzene are also efficiently activated, although complete conversion requires in those cases 1 hour.” and “Iodo heteroarenes such as 2-iodopyridine are also suitable substrates. Competitive nitrogen coordination slows down the process, but quantitative reaction is achieved over 3 days.”

8. The authors points out that it is key for the success of their procedure the presence of the coordinating N close to the gold center, which stabilizes the Au(III) adduct. Although Au(I) strongly prefers a 2-ligand coordination, it is also true that it can accept electronic density also in this oxidation state: do the authors observe any coordination by the NMe₂ group before the oxidative addition?

*No coordination of the NMe₂ group to gold(I) has been observed neither experimentally (no sign of N to Au interaction by NMR) nor theoretically (the DFT optimized structures of complexes **1-SbF₆** and **1-NTf₂** display long Au–N bond distances, $d_{\text{Au–N}} = 2.760\text{--}3.043$ Å, see Table S8 of the Supporting Information). According to calculations, the Au–N bond distance remains long even in the transition state for the oxidative addition (2.658–2.670 Å). Much shorter distances are observed in the oxidative addition products (2.344–2.350 Å).*

Minor points: *All the following minor points have been corrected. The recent work of Hashmi et al on C-C coupling of boronic acids with diazonium salts has been added as reference 28.*

- 1) page 2: there are 3 dots at the end of the first paragraph: I assume it is a typo
- 2) page 2: the last sentence of the second paragraph seems confusing to me and should be rephrased
- 3) caption of Scheme 1: something seems to be missing when reads: “Au-P [°]”

Reviewer #3

1. This manuscript describes results that expand upon previous work by this group on the oxidative addition of aryl iodides to gold(I) to form Au(III) complexes. What is new here is the simpler ligand design, a broader reaction scope and the demonstration that the resulting aryl-Au(III) complexes react with 1,3,5-trimethylbenzene both under stoichiometric and catalytic conditions. The observation of a room temperature oxidative addition of biphenylene to Au(I) is also interesting (ref. 31 should also have been cited in p. 5 within this context).

We surmise that the reviewer refers to ref 35 [ref 31 of the initial version, now reference 32, deals with reductive elimination processes from gold(III)]. This reference has been cited in p. 5 in the discussion of the oxidative addition of biphenylene.

2. However, although biaryls can be obtained from aryl iodides, this is not a “cross-coupling” as stated in the title of the manuscript and elsewhere, but rather an intermolecular arylation reaction. Cross-couplings are reactions between alkyl, alkenyl or aryl halides R'-X and organometallic reagents to form R-R'. Although this distinction might be considered by some as formal, formalities are important not to mislead the potential readers of this work.

We fully agree that semantic is important to precisely describe structures, transformations and phenomena. The reviewer argues that the cross-coupling terminology only applies to the coupling between organohalides and organometallic reagents. Although it is true that cross-coupling reactions were initially discovered and largely developed with organometallic reagents, the cross-coupling terminology intrinsically refers to reactions in which two organic fragments are coupled and thus also encompasses simple hydrocarbons. The Heck reaction is a very representative and famous example of a coupling reaction between an aryl halide and an alkene. Thus, we do believe the cross-coupling terminology is indeed suitable to describe the transformations studied in this work.

3. The authors state in the introduction that “Monocoordinate gold(I) cationic complexes [LAu]⁺ (L = R₃P or NHC) have also been shown to undergo oxidative addition (refs, 34, 41, 42)”. This statement is not correct (at least) for reactions proceeding in solution. In ref. 41, the group of Toste demonstrated that NHCAuCl reacts with biphenylene by oxidative addition in the presence of AgSbF₆ by forming an intermediate aryl complex, not via a “monocoordinate gold(I) cationic complexes [LAu]⁺”. Furthermore, in ref. 41 (see Figure 3), a “naked” [LAu]⁺ complex does not exist in the calculated reaction coordinate the oxidative addition. Finally, the evidence provided by the gas-phase study reported in ref. 42, actually indicates that if a “naked” [LAu]⁺ species would be generated in the gas-phase, its reaction with PhI would actually give first a two-coordinated [R₃PAu(PhI)]⁺ species.

It is clear that naked gold(I) cationic complexes of the general formula [LAu]⁺ do not exist in solution, the gold center interacting with the counteranion, a solvent molecule or any other electron donor. To avoid any misunderstanding, the statements referring to mono versus dicoordinate gold(I) cationic complexes have been modified as follows:

- “cationic gold(I) complexes featuring a small bite angle diphosphino-carborane ligand (DPCb) readily achieve the oxidative addition of Csp²-I and strained C-C bonds”
- “Gold(I) cationic complexes featuring monodentate ligands (L = R₃P or NHC) have also been shown to undergo oxidative addition”

4. Calculations are reported to support the relatively facile oxidative addition. Although those performed from the triflimide complex (Figure S49) are reasonable, the calculations shown in the

body of the manuscript start from LAu-SbF₆ complex, whose actual involvement as real species under the reaction conditions is rather unlikely. More probably, the intermediate [L-Au-IPh]⁺ complex is formed by an associative chloride substitution from L-AuCl assisted by Ag⁺. Although the involvement of an arene-gold(I) complex as intermediate species is considered in the SI (Figure S51), this is proposed to arise from totally unrealistic monocoordinated LAu⁺.

We agree that the different binding strengths of SbF₆⁻ and NTf₂⁻ to gold is important and it is actually what explains the difference in reactivity (cf the answer to reviewer 2, point 4). But we do not believe that the involvement of LAu-SbF₆ complexes is unlikely. Such species have actually been reported by Toste in the oxidative addition of biphenylene to NHC gold(I) complexes (ref 35). In addition, our calculations with the (P,N) ligand show that the LAu-SbF₆ complex and the corresponding I-adduct are very close in energy ($\Delta G = -3 \text{ kcal.mol}^{-1}$) and support their involvement. As mentioned above, the localization of the transition state for the displacement of SbF₆⁻ by PhI also completes the picture and confirms that this step proceeds with a very low activation barrier.

5. In addition, although experts in the field might understand that Au(I) is actually bound to the halogen in LAu-SbF₆ and other complexes shown in Figure 1 with SbF₆ or GaCl₄ as the ligands, the depiction chosen by these authors for the bonding is, at least, very misleading.

We used in Figure 1 the depiction inorganic chemists classically use for ion pair complexes. We agree it is simplified as it does not precise the nature of the cation/anion interaction and which atoms are involved. To avoid any misunderstanding, the formula of the counteranions (SbF₆, GaCl₄...) have been developed to explicitly depict the contact between the gold center and one halogen atom. The same depiction has been used in Figure 2 for consistency.

6. Supporting information: In general, yields given for the described new complexes are based on NMR and not on the isolated pure compounds. Furthermore, elemental analyses were not reported for any of the gold(III) complexes, which is below the standards required for characterization of organometallic compounds. Three of the complexes resulting from oxidative addition of aryl bromides reported in p. S12 were only characterized by ³¹P NMR and HRMS.

It is important to recall here that oxidative addition of aryl halides to gold remains very rare and that the resulting cationic gold(III) species are quite sensitive and reactive compounds. Combined with their tendency for I and Br to Cl exchange, this makes the isolation of pure [(P,N)AuArX]⁺SbF₆⁻ complexes very challenging. Since the purpose of the study was not to isolate a series of such complexes, but to demonstrate their ready formation by oxidative addition and subsequent reactivity in cross-coupling, most gold(III) complexes have been essentially characterized in situ by NMR and mass spectrometry (HRMS). Nonetheless, the possibility to isolate such species has been demonstrated (cf compound 2' obtained by reacting 2 with nBu₄NCl).

7. The same lack of experimental rigor applies for the formation of the biaryls. In all cases, yields of 2,4,6-trimethoxy-1,1'-biphenyl were determined by GC. For the other biphenyls described in pp. S18-19, it is not clear if the yields correspond to isolated materials. All the melting points for these biaryls are missing. Biphenyl 22 is reported as a new compound, although it has been reported: Tetrahedron Lett. 2014, 55, 3184; 2016, 57, 4235. Melting point and HRMS (or elemental analysis) are missing for this compound. For the two diphenyl-1H-pyrroles, which were obtained as a mixture, only ¹³C NMR spectra are reported. The ³¹P NMR spectra of 2, 1, 13, 14 shows another minor signal. The ¹H NMR spectra of 3, 4, 5, 6, 8, 9, 10, 11, 12 show a large amount of ArI and/or other impurities. Only

two melting points are actually reported in this work (for complexes 13, 14). The statement that melting points are uncorrected is not correct. See: Tiers, G. J. Chem. Educ. 1990, 67, 258.

Minor corrections: 1-phenyl-pyrrole should be 1-phenylpyrrole.

All the biaryl products deriving from the cross-coupling of trimethoxybenzene with aryl iodides and bromides were isolated in pure forms by column chromatography and the yields given in page S18-19 are isolated yields. We agree it was not clear in the general procedure described in page S17. We apologize for that and have modified it accordingly.

The structures of the biaryl products were unambiguously authenticated by comparing their NMR signatures with those reported in the literature. We thank the reviewer for pointing out that the product 22 was described in two Tetrahedron Letters papers. These references have been added in the Supporting Information.

The ¹H NMR spectrum of the coupling product between 1-phenylpyrrole and PhI was provided (Figure S45).

For complexes 3, 4, 5, 6, 8, 9, 10, 11, 12, we have provided in the Supporting Information, the ¹H NMR spectra of the crude reaction mixtures. Excess ArI and small impurities can be detected, but we think these spectra are more representative of the oxidative addition reactions and indeed substantiate the quite clean formation of the gold(III) species.

The statement about the melting point determination in the general description of the materials and methods (page S2) has been modified as follows: "Melting points were determined with a calibrated Stuart SMP40 (PT1000) apparatus."

1-phenyl-pyrrole has been changed for 1-phenylpyrrole in the main text and Supporting Information, as suggested.

8. The weakest point of this study is that scope of the C-C bond formation is essentially limited to 1,3,5-trimethylbenzene as the electron-rich partner. In addition, the quality of the experimental data is well below the standards required for publication of organometallic and/or organics chemistry work in specialized journals such as Organometallics, Dalton, or Org. Lett. As indicated above, this work contains sufficient elements of novelty to be published. However, considering all the precedents reported by this and other groups, the rather preliminary nature of the calculations, and the poor quality of the characterization data, publication of this work in Nature Communications is not recommended.

We have the feeling that the reviewer somewhat minimizes the scope of our study, not considering the ability of the (P,N) gold complex to activate and couple Ar-Br substrates and the arylation of 1-phenylpyrrole, which shows that the gold-catalyzed direct arylation process has some generality.

We do not agree with the reviewer about the quality of the experimental part, the rigor of the study and the preliminary nature of the DFT calculations. We do think the Supporting Information meets the requirements of Nature Communications in terms of analytical data and that the work has been achieved at very good technical standard.

According to reviewer 1, "...the characterization of the new compounds is exhaustive and I appreciate that NMR spectra are included in the Supporting Information. DFT studies are very informative about the core subject..."

REVIEWERS' COMMENTS:

Reviewer #1 (Remarks to the Author):

The authors have addressed all the questions and concerns raised by this and other referees. I am fully content with the updated version of the manuscript and once more I certify my most enthusiastic support for the final acceptance of this work.

Reviewer #2 (Remarks to the Author):

I am satisfied with the revisions and clarifications provided by the authors.

Reviewer #3 (Remarks to the Author):

Regarding the discussion on semantics, in the opinion of this reviewer, the term cross-coupling should be used restrictively. Actually, the Heck reaction is not a "very representative and famous example of a coupling reaction". This famous reaction is an alkenylation reaction, not a cross-coupling. The Heck reaction shares with cross couplings the first part of the catalytic cycle, but the second part is fundamentally different as it involves a migratory insertion, not a transmetalation, which is a mechanistically quite different transformation. I would still recommend the authors to use the term "arylation" or ("intermolecular arylation") in the title and elsewhere to avoid adding confusion to this field.

The authors argue that "LAu-SbF₆ species have actually been reported by Toste in the oxidative addition of biphenylene to NHC gold(I) complexes (ref 35)". However, in that report and in many others, what is described is the in situ reaction of LAuCl with AgSbF₆ (see the supporting information in Toste's paper, preparation of 3 from 1). The presumed "LAu-SbF₆" complex might have been formed, but it was not isolated or characterized in solution. What is actually well known is the formation of chloride-bridged dinuclear complexes of the type [LAuClAuL]X (see: Homs, A.; Escofet, I.; Echavarren, A. M. *Org. Lett.* 2013, 15, 5782).

The authors correctly argue in their response to the referees that the work reported in two of the new references that have been added (refs. 49 and 50) were actually disputed by another group. Actually, the catalytic results described in ref. 49 were reinterpreted by the original authors as resulting from nanoclusters. None of these two references report "well-defined mononuclear gold complexes" as stated at the end of the last paragraph of the revised version of the manuscript. This is simply not correct: in ref. 50 the "gold complex" was made in situ from AuI and dppf, under conditions which would presumably give rise to a dinuclear gold(I) complex. On the other hand, the gold(I) claimed to be the active catalyst in ref. 49, was drawn in the original publication as a trinuclear gold complex.

As concluded in my first report, this work contains sufficient elements of novelty to be published. Now, in the revised version, a few questions of the calculations have been clarified and the experimental characterization data has been slightly improved. I understand that the main point of this work is the demonstration that the oxidative addition of aryl halides can be coupled with the reaction of the resulting gold(III) complexes electron-rich aromatic compounds, in what it is essentially an electrophilic aromatic reaction. However, whereas the scope for the oxidative addition is good, only two electron-rich aromatic compounds (1,3,5-trimethoxybenzene and pyrrole) have been successfully used for the second arylation step and in the second case, two products are formed. This narrow scope (for the last reaction) should be specifically indicated in the conclusions.

All in all, I would support publication of a revised version of this manuscript once all the above-mentioned issues have been addressed by the authors.

Point-by-point response to the referee's comment

Reviewer #3

1. Regarding the discussion on semantics, in the opinion of this reviewer, the term cross-coupling should be used restrictively. Actually, the Heck reaction is not a “very representative and famous example of a coupling reaction”. This famous reaction is an alkenylation reaction, not a cross-coupling. The Heck reaction shares with cross couplings the first part of the catalytic cycle, but the second part is fundamentally different as it involves a migratory insertion, not a transmetalation, which is a mechanistically quite different transformation. I would still recommend the authors to use the term “arylation” or (“intermolecular arylation”) in the title and elsewhere to avoid adding confusion to this field.

Again, we do not agree with the reviewer's view and consider his/her definition of cross-coupling reactions as personal. There is no reason to restrict the terminology to reactions combining R-X and R-M reagents. As a matter of fact, the Nobel prize in Chemistry was awarded in 2010 jointly to Richard F. Heck, Ei-ichi Negishi and Akira Suzuki "for palladium-catalyzed cross couplings in organic synthesis". This said, we have no problem to use the term arylation instead of cross-coupling and have modified the title and manuscript accordingly.

2. The authors argue that “LAu-SbF₆ species have actually been reported by Toste in the oxidative addition of biphenylene to NHC gold(I) complexes (ref 35)”. However, in that report and in many others, what is described is the in situ reaction of LAuCl with AgSbF₆ (see the supporting information in Toste's paper, preparation of 3 from 1). The presumed “LAu-SbF₆” complex might have been formed, but it was not isolated or characterized in solution. What is actually well known is the formation of chloride-bridged dinuclear complexes of the type [LAuClAuL]X (see: Homs, A.; Escofet, I.; Echavarren, A. M. *Org. Lett.* 2013, 15, 5782).

LAu/SbF₆ species have indeed not been isolated but Toste et al report in their Nature paper their generation and NMR characterization (after removal of the silver salt upon filtration over celite). In the absence of XRD data, one may argue about their exact structure. But anyhow and as mentioned previously, our calculations with the (P,N) ligand show that the LAu/SbF₆ complex and the corresponding I-adduct are very close in energy and likely involved in the oxidative addition reaction.

3. The authors correctly argue in their response to the referees that the work reported in two of the new references that have been added (refs. 49 and 50) were actually disputed by another group. Actually, the catalytic results described in ref. 49 were reinterpreted by the original authors as resulting from nanoclusters. None of these two references report “well-defined mononuclear gold complexes” as stated at the end of the last paragraph of the revised version of the manuscript. This is simply not correct: in ref. 50 the “gold complex” was made in situ from AuI and dppf, under conditions which would presumably give rise to a dinuclear gold(I) complex. On the other hand, the gold(I) claimed to be the active catalyst in ref. 49, was drawn in the original publication as a trinuclear gold complex.

The citation of references 49-51 without footnote was indeed confusing. To avoid any misunderstanding, the main text has been modified as follows: “The potential of (P,N) gold(I) complexes in Au(I)/Au(III) catalysis is also demonstrated by the development of the first arylation

reactions involving aryl halides and catalyzed by well-defined mononuclear gold complexes. Precedents of gold-catalyzed cross-coupling reactions with aryl halides are extremely rare,⁴⁹⁻⁵¹ and involve polynuclear species.”

4. As concluded in my first report, this work contains sufficient elements of novelty to be published. Now, in the revised version, a few questions of the calculations have been clarified and the experimental characterization data has been slightly improved. I understand that the main point of this work is the demonstration that the oxidative addition of aryl halides can be coupled with the reaction of the resulting gold(III) complexes electron-rich aromatic compounds, in what it is essentially an electrophilic aromatic reaction. However, whereas the scope for the oxidative addition is good, only two electron-rich aromatic compounds (1,3,5-trimethoxybenzene and pyrrole) have been successfully used for the second arylation step and in the second case, two products are formed. This narrow scope (for the last reaction) should be specifically indicated in the conclusions. All in all, I would support publication of a revised version of this manuscript once all the above-mentioned issues have been addressed by the authors.

Accordingly, the conclusion has been modified as follows: “Future work will seek to generalize this ligand design principle to other classes of bidentate ligands, to extend the scope of electron-rich (hetero)arenes and to develop further gold-catalyzed cross-coupling reactions with aryl halides.”.